# Postpartum hemorrhage in Suriname: A national descriptive study of hospital births and an audit of case management

**Lachmi R. Kodan**[1,2]*, **Kim J. C. Verschueren**[2], **Zita D. Prüst**[2], **Nicolaas P. A. Zuithoff**[3], **Marcus J. Rijken**[2,3], **Joyce L. Browne**[3], **Kerstin Klipstein-Grobusch**[3,4], **Kitty W. M. Bloemenkamp**[2], **Antoon W. Grunberg**[5]

1 Department of Obstetrics and Gynecology, Academic Hospital Paramaribo, Paramaribo, Suriname, South Africa, 2 Division Women and Baby, Department of Obstetrics, Birth Centre Wilhelmina's Children Hospital, University Medical Center Utrecht, Utrecht University, Utrecht, The Netherlands, 3 Julius Global Health, Julius Center for Health Sciences and Primary Care, University Medical Center Utrecht, Utrecht University, Utrecht, The Netherlands, 4 Division of Epidemiology and Biostatistics, School of Public Health, Faculty of Health Sciences, University of the Witwatersrand, Johannesburg, South Africa, 5 Board of Doctoral Graduations and Honorary Doctorate Awards, Anton de Kom University, Paramaribo, Suriname, South Africa

* lachmikodan@yahoo.com

**Data Availability Statement:** Although de-identified, the data contains identifying or sensitive patient information that cannot be shared publicly.

## Abstract

### Background

Postpartum hemorrhage (PPH) is the leading cause of direct maternal mortality globally and in Suriname. We aimed to study the prevalence, risk indicators, causes, and management of PPH to identify opportunities for PPH reduction.

### Methods

A nationwide retrospective descriptive study of all hospital deliveries in Suriname in 2017 was performed. Logistic regression analysis was applied to identify risk indicators for PPH ($\geq$ 500ml blood loss). Management of severe PPH (blood loss $\geq$1,000ml or $\geq$500ml with hypotension or at least three transfusions) was evaluated via a criteria-based audit using the national guideline.

### Results

In 2017, the prevalence of PPH and severe PPH in Suriname was 9.2% (n = 808/8,747) and 2.5% (n = 220/8,747), respectively. PPH varied from 5.8% to 15.8% across the hospitals. Risk indicators associated with severe PPH included being of African descent (Maroon aOR 2.1[95%CI 1.3–3.3], Creole aOR 1.8[95%CI 1.1–3.0]), multiple pregnancy (aOR 3.4[95%CI 1.7–7.1]), delivery in Hospital D (aOR 2.4[95%CI 1.7–3.4]), cesarean section (aOR 3.9[95% CI 2.9–5.3]), stillbirth (aOR 6.4 [95%CI 3.4–12.2]), preterm birth (aOR 2.1[95%CI 1.3–3.2]), and macrosomia (aOR 2.8 [95%CI 1.5–5.0]). Uterine atony (56.7%, n = 102/180[missing 40]) and retained placenta (19.4%, n = 35/180[missing 40]), were the main causes of severe PPH. A criteria-based audit revealed that women with severe PPH received prophylactic oxytocin in 61.3% (n = 95/155[missing 65]), oxytocin treatment in 68.8% (n = 106/154[missing 66]), and tranexamic acid in 4.9% (n = 5/103[missing 117]).

Data from this study can be requested from the ethical review board of the Surinamese Committee on Research Involving Human Subjects at cmwohealth@gmail.com or vogezdir@gmail.com.

**Funding:** The author(s) received no specific funding for this work.

**Competing interests:** The authors have declared that no competing interests exist.

## Conclusions

PPH prevalence and risk indicators in Suriname were similar to international and regional reports. Inconsistent blood loss measurement, varied maternal and perinatal characteristics, and variable guideline adherence contributed to interhospital prevalence variation. PPH reduction in Suriname can be achieved through prevention by practicing active management of the third stage of labor in every birth and considering risk factors, early recognition by objective and consistent blood loss measurement, and prompt treatment by adequate administration of oxytocin and tranexamic acid according to national guidelines.

## Introduction

Postpartum hemorrhage (PPH) remains the most frequent cause of maternal mortality, accounting for 27% of maternal deaths worldwide [1]. Most of these deaths occur in low- and middle-income countries (LMIC) and are associated with limited access to timely and quality care and inadequate availability of resources such as blood products [2, 3]. PPH has become more prevalent due to increasing rates of advanced maternal age, obesity, preeclampsia, prolonged labor, cesarean delivery, and multiple pregnancies [4–7]. Besides, PPH contributes to severe maternal morbidity and permanent disability worldwide [8]. Global PPH prevalence ranges from 6 to 10% but varies widely between and within countries [9–11]. In Latin America and the Caribbean (LAC), the estimated prevalence of PPH is between 8.2% and 8.9%, and severe PPH (defined as blood loss ≥ 1,000 ml) occurred between 3.3% and 5.3% of births [10, 11].

The main causes of PPH are the "4 T's": uterine atony (tone, 80%), genital tract laceration (trauma, 13%), retained placenta or placental tissue (tissue, 5%), and coagulopathy (thrombin, 2%) [8, 12, 13]. While risk indicators are associated with various socio-demographics, pregnancy complications, and delivery characteristics, many women experience PPH without exhibiting any specific risk indicator [12, 14, 15]. Therefore, prevention, early recognition, and prompt PPH treatment for each woman remain the cornerstone to avoid maternal morbidity and mortality [13, 16, 17].

In Suriname, PPH was the leading direct cause of maternal mortality responsible for 20% (n = 13/65) of deaths from 2010 to 2014. Delays in diagnosis, monitoring, and treatment were critical factors contributing to these deaths [17]. A national PPH guideline developed in 2016 incorporates international recommendations for prevention (screening for and treating anemia and active management of the third stage of labor (AMTSL)), early recognition (measurement or visual estimation of the amount of blood loss and clinical signs), and management (oxytocin prevention and therapy and tranexamic acid use) [18]. However, no detailed information on PPH prevalence, causes, and risk indicators were available for Suriname. Therefore, this study aimed to (1) assess the prevalence of PPH, (2) identify risk indicators and underlying causes of PPH, and (3) evaluate the management of severe PPH by performing a criteria-based audit. Specific gaps identified provide evidence to guide further efforts to reduce PPH-related maternal mortality and morbidity.

## Methods

### Study design and setting

A nationwide retrospective descriptive study of all hospital deliveries was conducted in Suriname between January 1st and December 31st, 2017. In addition, a criteria-based audit was performed to analyze case management of severe PPH.

Suriname is a middle-income country on the northern coast of South America with the lowest population density on the continent. More than 80% of the estimated population of 583,200 lives in urban and rural coastal lowlands [19]. The ethnic distribution includes Hindustani (27%), Maroon (22%), Creole (16%), Javanese (14%), mixed (combination of ethnicities– 13%), Indigenous (4%), and others (Chinese, Brazilian, Caucasian, and unknown– 4%) [20]. Maroons and Creoles are of African ancestry, while Hindustani and Javanese are of Asian descent. Of the approximately 10,000 deliveries per year, 92% are institutional (86% hospital, 6% primary care) [20]. Four out of five major hospitals are in the capital Paramaribo; one is located at the western border of Suriname (Nickerie). All complicated pregnancies and births in primary care, including women with ongoing or severe PPH, are transferred to the nearest hospital. Every hospital has an intensive care unit (ICU).

### Data collection and variables

Birth attendants documented each birth with gestational age $\geq$ 22 weeks and birth weight of $\geq$ 500 grams in a parturition book. The blood loss amount was usually visually estimated. In the case of estimated high blood loss, blood in the bedpan was measured with a measuring jug (mL) or the pads were weighted (grams = mL). However, in two hospitals, only blood clots were measured [18]. Hospital administrative personnel anonymously entered data from the paper parturition books into a password-secured digital database on a daily basis. The datasets from the five hospitals were merged, yielding one national delivery database for 2017. Missing and incorrect data were crosschecked with the original parturition books and medical files. The Surinamese Obstetric Surveillance System (SurOSS) identified all women with potentially life-threatening disorders in pregnancy between March 2017 and February 2018 [21]. Study data for the criteria-based audit were derived from this database.

The primary outcome variable of our study was PPH, which was defined as a blood loss of at least 500 ml within 24 hours postpartum. Moderate PPH was defined as blood loss between 500 and 999 ml. Severe PPH was defined as blood loss of at least 1,000 ml, bleeding associated with hypotension (systolic blood pressure below 90 mm Hg with a pulse rate higher than 90 beats per minute), or transfusion of at least three units of blood products based on the criteria of SurOSS [21]. The available independent variables (maternal, pregnancy, and delivery characteristics) were categorized according to international classifications (S1 Fig in S1 File). The criteria-based audit was confined to severe PPH. Prevention and management of severe PPH were audited using the national PPH guideline [18]. Detailed information on the cause, course, and management of severe PPH was not always available (S2 Fig in S1 File). This manuscript was written in accordance with the Strengthening the Reporting of Observational Studies in Epidemiology Guidelines [22].

### Statistical analysis

Data were analyzed using IBM SPSS version 24.0 (Armonk, New York, USA) and SAS v9.4 (SAS Institute, Gary Indiana). Frequencies of maternal, delivery and perinatal characteristics were calculated in women with and without PPH. Logistic regression was used to investigate the independent association of risk indicators with moderate and severe PPH. Univariate regression analysis generated odds ratios (ORs) with 95% confidence interval (CI). The multivariable regression analysis included all variables with a p-value < 0.10 from the univariate analysis and variables reported by the literature as important risk indicators (e.g., multiple gestations, parity). These were presented as adjusted OR (aOR) with 95% CI. A Pareto chart was used to prioritize areas for the quality of care improvement, applying the "80–20 rule" of the Pareto principle, which suggests that most problems (80%) are due to a few key causes (20%)

[23]. Clinical management of PPH was reported as frequencies and percentages after applying the audit criteria. Pearson correlation was used to evaluate the association between blood loss, units of blood transfused, and ICU admission.

### Ethical considerations

This research was performed according to the Declaration of Helsinki. The ethical review board of the Surinamese Committee on Research Involving Human Subjects approved the study on maternal morbidity on October 4[th], 2016 (VG21-16) and the study on postpartum hemorrhage on September 8[th], 2018 (VG11-18). The registry data was anonymous and aggregated, and the need for individual consent was waived.

### Results

Blood loss was documented in 96.4% (n = 8,747/9,071) of the hospital deliveries in 2017 (Table 1). The median blood loss of all included women who gave birth was 150 ml (range 0–4,620). PPH occurred in 9.2% (n = 808/8,747) of the deliveries, with 6.7% (n = 588/8,747) being moderate and 2.5% (n = 220/8,747) severe PPH. The diagnosis of severe PPH was based on blood loss of more than 1,000 ml in 82.7% (n = 182/220) of women, and in 17.3% (n = 38/220) blood loss was moderate, but at least three units of blood products were transfused, or there was hemodynamic instability. In Table 1, the maternal, perinatal and delivery characteristics of the births with and without PPH are compared. Pre-delivery anemia occurred in 34.9% (n = 65/186 [missing 622]) of women with PPH. Women of African descent were more frequently anemic antepartum (63.0%, n = 677/1,074) than women from other ethnicities were (based on data availability of only two hospitals). The prevalence of PPH was higher in women delivered by cesarean section (CS) than those delivered by a vaginal birth (20.8%, n = 400/1,924 vs 6.0%, n = 408/6,823, respectively, p<0.01). There were nine maternal deaths, three of which were complicated by PPH. Information on blood loss was most frequently missing in CS (65.7%, n = 213/324), low birth weight (29.1%, n = 94/324), and preterm births (25.7%, n = 81/324) (Table 1).

The prevalence of moderate and severe PPH in the hospitals varied significantly between 4.5 to 11.8% (p<0.001) and 1.3 to 4.0% (p<0.001), respectively, with the highest prevalence in Hospital D and the lowest in Hospital E (Fig 1). CS prevalence was highest in Hospitals D (24.6%, n = 604/2,456) and E (37.8%, n = 564/1,493) and lowest in Hospital C (14.4%, n = 53/367) (S1 Table in S1 File). PPH after CS was more common in Hospital D than in other hospitals (48.4% vs 6.9–13.9%, p<0.001). In Hospitals A, B, and D, women giving birth were more often of African descent (68.0% vs 51.8% vs 55.5%) compared to Hospitals C (5.8%) and E (26.6%). AMTSL (by administration of oxytocin for PPH prevention) was applied less frequently for severe PPH cases in Hospital D than in the other hospitals (46%, n = 29/155 [missing 65] vs 67.9–77.8%) (S1 Table in S1 File).

The logistic regression analysis for moderate and severe PPH is presented in Table 2. Women of Creole and Maroon ethnicity had significantly higher odds of developing severe PPH than Hindustani women did (aOR 1.8 [95%CI 1.1–3.0] vs 2.1 [95%CI 1.3–3.3], respectively). Women delivering in Hospital D were more likely to experience moderate (aOR 2.7 [95%CI 2.2–3.4]) and severe PPH (aOR 2.4 [95%CI 1.7–3.4]) compared to Hospital B. Also, the risk of both moderate and severe PPH was significantly higher in women delivering by CS (aOR 5.4 [95%CI 4.5–6.6] vs aOR 3.9 [95%CI 2.9–5.3]) compared to vaginal delivery. Other strongly associated risk indicators for severe PPH were stillbirths (aOR 6.4 [95%CI 3.4–12.2]), multiple pregnancy (aOR 3.4 [95%CI 1.7–7.1]), very preterm birth (aOR 2.3 [95%CI 1.1–4.9]), preterm birth (aOR 2.1 [95%CI 1.3–3.2]), and neonatal macrosomia (aOR 2.8 [95%CI 1.5–

**Table 1. Maternal, perinatal, and delivery characteristics of births in Suriname in 2017 with and without postpartum hemorrhage and undocumented blood loss.**

| | No PPH n (%) | Moderate PPH[1] n (%) | Severe PPH[2] n (%) | Undocumented blood loss n (%) |
|---|---|---|---|---|
| Total | 7,939 (100%) | 588 (100%) | 220 (100%) | 324 (100%) |
| Live births | 7,811 (98.4) | 577 (98.1) | 200 (91.9) | 301 (92.9) |
| Stillbirths | 128 (1.6) | 11 (1.9) | 20 (9.1) | 23 (7.1) |
| Maternal characteristics | | | | |
| Age (years) | | | | |
| 12–19 | 1,164 (14.7) | 64 (10.9) | 24 (10.9) | 37 (11.5) |
| 20–34 | 5,678 (71.6) | 421 (71.6) | 147 (66.8) | 228 (70.8) |
| $\geq$ 35 | 1,087 (13.7) | 103 (17.5) | 49 (22.3) | 57 (17.7) |
| Missing | 10 | 0 | 0 | 2 |
| Ethnicity | | | | |
| Maroon | 2,271 (28.8) | 151 (25.9) | 80 (36.4) | 82 (26.0) |
| Creole | 1,709 (21.7) | 124 (21.2) | 52 (23.6) | 85 (27.0) |
| Hindustani | 1,469 (18.6) | 88 (15.0) | 27 (12.3) | 71 (22.5) |
| Other[3] | 1,301 (16.5) | 118 (20.2) | 29 (13.2) | 46 (14.6) |
| Javanese | 835 (10.6) | 76 (13) | 21 (9.5) | 22 (7.0) |
| Indigenous | 295 (3.7) | 27 (4.6) | 11 (5.0) | 9 (2.9) |
| Missing | 59 | 4 | 0 | 9 |
| Maternal HIV status | | | | |
| Positive | 58 (0.8) | 5 (0.9) | 2 (1.0) | 3 (0.9) |
| Missing | 333 | 24 | 11 | 0 |
| Pregnancy characteristics | | | | |
| Parity | | | | |
| 0 | 2,729 (34.5) | 205 (35) | 65 (29.5) | 113 (36.1) |
| 1–4 | 4,579 (57.8) | 334 (57) | 125 (56.8) | 174 (55.6) |
| $\geq$ 5 | 612 (7.7) | 47 (8) | 30 (13.6) | 26 (8.3) |
| Missing | 19 | 2 | 0 | 11 |
| Gestational age | | | | |
| < 32 weeks | 231 (2.9) | 12 (2) | 17 (7.8) | 34 (10.8) |
| 32–36 weeks | 806 (10.2) | 79 (13.5) | 45 (20.5) | 47 (14.9) |
| $\geq$ 37 weeks | 6,863(86.4) | 495 (84.5) | 157 (71.7) | 234 (74.3) |
| Missing | 39 | 2 | 1 | 9 |
| Antepartum anemia | | | | |
| Anemia[4] | 990 (37.6) | 46 (34.3) | 19 (36.5) | 30 (36.6) |
| Missing | 5,307 | 454 | 168 | 242 |
| Type of pregnancy | | | | |
| Multiple pregnancy | 88 (1.1) | 9 (1.5) | 10 (4.5) | 9 (2.8) |
| Delivery characteristics | | | | |
| Hospital of delivery | | | | |
| A | 1,884 (23.7) | 91 (15.5) | 44 (20.0) | 82 (25.3) |
| B | 2,379 (30.0) | 130 (22.1) | 53 (24.1) | 92 (28.4) |
| C | 333 (4.2) | 24 (4.1) | 10 (4.5) | 24 (4.1) |
| D | 1,977 (24.9) | 277 (47.1) | 94 (42.7) | 108 (33.3) |
| E | 1,366 (17.2) | 66 (11.2) | 19 (8.6) | 42 (13.0) |
| Onset of labor | | | | |
| Augmentation | 1,771 (47.4) | 86 (45.7) | 38 (48.1) | 32 (33.0) |
| Missing | 4,202 | 400 | 141 | 227 |
| Mode of delivery | | | | |

*(Continued)*

**Table 1.** (Continued)

|  | *No PPH n (%)* | *Moderate PPH[1] n (%)* | *Severe PPH[2] n (%)* | *Undocumented blood loss n (%)* |
|---|---|---|---|---|
| Spontaneous | 6,289 (79.2) | 267 (45.4) | 127 (57.7) | 107 (33) |
| Cesarean section | 1,524 (19.2) | 311 (52.9) | 89 (40.5) | 213 (65.7) |
| Instrumental | 126 (1.6) | 10 (1.7) | 4 (1.8) | 4 (1.2) |
| Vaginal laceration |  |  |  |  |
| 2nd grade or higher | 1,644 (50.6) | 86 (55.8) | 26 (41.9) | 19 (29.7) |
| Missing | 4,961 | 434 | 158 | 260 |
| Birthweight (grams) |  |  |  |  |
| < 2,500 | 1,115 (14.1) | 69 (11.7) | 48 (22.2) | 94 (29.1) |
| 2,500–3,999 | 6,582 (83.2) | 483 (82.6) | 154 (71.3) | 219 (67.8) |
| ≥ 4,000 | 211 (2.7) | 33 (5.6) | 14 (6.5) | 10 (3.1) |
| Missing | 31 | 3 | 4 | 1 |

[1] Blood loss 500–999 ml.

[2] Blood loss ≥ 1,000 ml or blood loss < 1,000 ml with hemodynamic instability or three or more units blood transfusion.

[3] Ethnicity other: Mixed, Chinese, Brazilian, Caucasian, or unknown.

[4] Hemoglobin ≤ 100 g/l or 6.1 mmol/l.

5.0]). At least one risk indicator was present in 70.1% (n = 6,130/8,747) of the births without PPH and in 80.8% (n = 653/808) of births complicated by PPH.

The Pareto chart shows that uterine atony (56.7%, n = 102/180[missing 40]) and retained placenta (19.4%, n = 35/180[missing 40]) caused almost 80% of severe PPH (Fig 2). Severe PPH occurred among women with preeclampsia in 23.2% (n = 45/194 [missing 26]) and

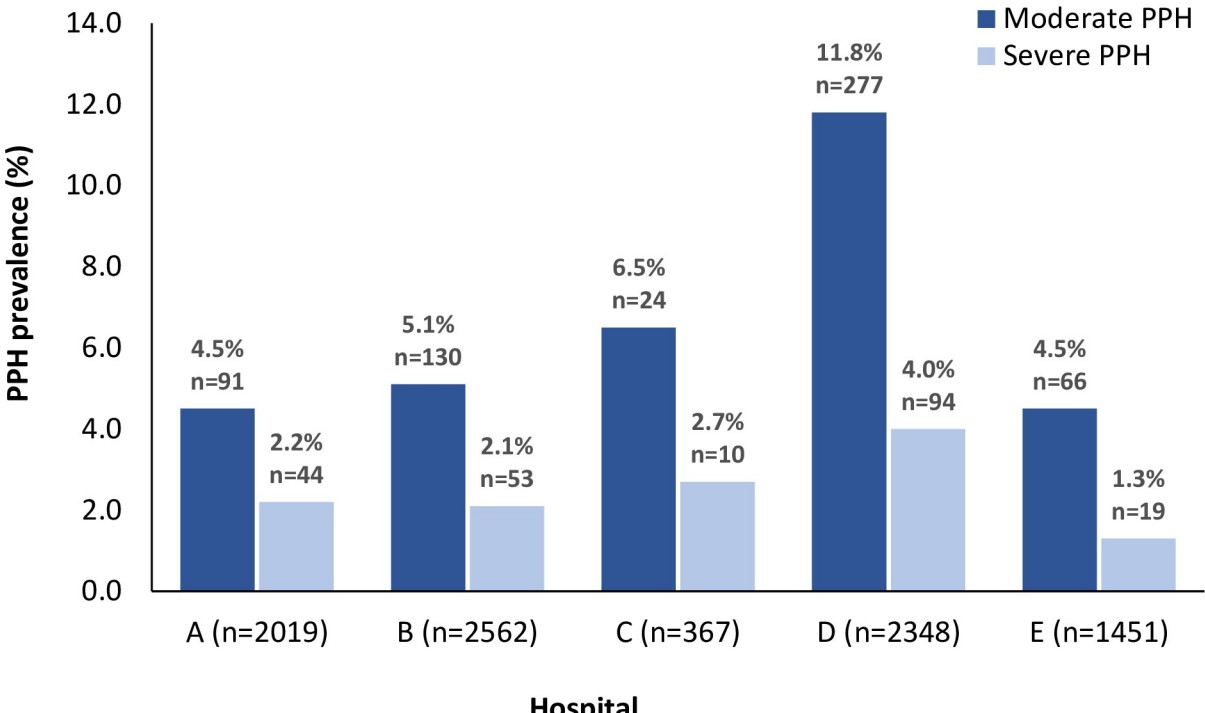

**Fig 1. Prevalence of moderate and severe postpartum hemorrhage (PPH) per hospital in Suriname in 2017.**

**Table 2. Univariate and multivariate logistic regression for moderate and severe postpartum hemorrhage (PPH).**

| | Moderate PPH[1] | | Severe PPH[2] | |
| --- | --- | --- | --- | --- |
| | Unadjusted Odds ratio (95% CI) | Adjusted[3] Odds ratio (95% CI) | Unadjusted Odds ratio (95% CI) | Adjusted[3] Odds ratio (95% CI) |
| | | p = 0.003 | p < 0.001 | p < 0.001 |
| Live birth | Reference | Reference | Reference | Reference |
| Stillbirth | 1.2 (0.6–2.2) | **2.9 (1.4–6.0)** | **6.1 (3.7–10.0)** | **6.4 (3.4–12.2)** |
| Maternal characteristics | | | | |
| Age (years) | p = 0.004 | | p = 0.001 | |
| 12–19 | 0.7 (0.6–1.0) | 0.8 (0.6–1.1) | 0.8 (0.5–1.2) | 0.8 (0.5–1.3) |
| 20–34 | Reference | Reference | Reference | Reference |
| $\geq$ 35 | **1.3 (1.2–1.7)** | 1.1 (0.9–1.4) | **1.7 (1.3–2.4)** | 1.4 (0.9–2.0) |
| Ethnicity | p = 0.02 | p = 0.001 | p = 0.04 | p = 0.07 |
| Maroon | 1.1 (0.9–1.5) | **1.5 (1.1–2.0)** | **1.9 (1.2–3.0)** | **2.1 (1.3–3.3)** |
| Creole | 1.2 (0.9–1.6) | **1.5 (1.1–2.0)** | 1.7 (1.0–2.7) | **1.8 (1.1–3.0)** |
| Hindustani | Reference | Reference | Reference | Reference |
| Other[4] | 1.5 (1.1–2.0) | **1.8 (1.3–2.4)** | 1.2 (0.7–2.1) | 1.3 (0.8–2.3) |
| Javanese | 1.5 (1.1–2.1) | **2.0 (1.4–2.8)** | 1.4 (0.8–2.4) | 1.9 (1.0–3.4) |
| Indigenous | 1.5 (1.0–2.4) | 1.5 (0.9–2.5) | 2.0 (1.0–4.1) | 2.0 (0.9–4.1) |
| Maternal HIV status | | | | |
| Positive | 1.2 (0.5–2.9) | | 1.3 (0.3–5.2) | |
| Negative | Reference | | Reference | |
| Pregnancy characteristics | | | | |
| Parity | | | p = 0.005 | |
| 0 | 1.0 (0.7–1.2) | 1.1 (0.9–1.3) | 0.9 (0.6–1.2) | 1.1 (0.8–1.5) |
| 1–4 | Reference | Reference | Reference | Reference |
| $\geq$ 5 | 1.1 (0.8–1.5) | 1.1 (0.8–1.6) | **1.8 (1.2–2.7)** | 1.2 (0.8–1.9) |
| Gestational age | p = 0.03 | p = 0.05 | p < 0.001 | p = 0.004 |
| < 32 weeks | 0.7 (0.4–1.3) | 1.2 (0.6–2.4) | **3.2 (1.9–5.4)** | **2.3 (1.1–4.9)** |
| 32–36 weeks | **1.4 (1.1–1.7)** | **1.5 (1.1–2.0)** | **2.4 (1.7–3.4)** | **2.1 (1.3–3.2)** |
| $\geq$ 37 weeks | Reference | Reference | Reference | Reference |
| Antepartum anemia | | | | |
| No anemia | Reference | | Reference | |
| Anemia[5] | 0.9 (0.6–1.3) | | 1.0 (0.5–1.8) | |
| Type of pregnancy | | | p < 0.001 | p < 0.001 |
| Singleton | Reference | Reference | Reference | Reference |
| Multiple | 1.4 (0.7–2.8) | 1.4 (0.6–3.0) | **4.3 (2.2–8.3)** | **3.4 (1.7–7.1)** |
| Delivery characteristics | | | | |
| Hospitals | p < 0.001 | p < 0.001 | p < 0.001 | p < 0.001 |
| A | 0.9 (0.7–1.2) | 1.0 (0.7–1.3) | 1.1 (0.7–1.6) | 1.0 (0.6–1.5) |
| B | Reference | Reference | Reference | Reference |
| C | 1.3 (0.8–2.1) | 1.7 (1.0–2.8) | 1.4 (0.7–2.7) | 2.1 (1.0–4.4) |
| D | **2.6 (2.1–3.2)** | **2.7 (2.2–3.4)** | **2.1 (1.5–3.0)** | **2.4 (1.7–3.4)** |
| E | 0.9 (0.7–1.2) | 0.6 (0.4–0.8) | 0.6 (0.4–1.1) | 0.5 (0.3–0.9) |
| Onset of labor | | | | |
| Spontaneous | Reference | | Reference | |
| Augmentation | 0.9 (0.7–1.3) | | 1.0 (0.7–1.6) | |
| Mode of delivery | p < 0.001 | p < 0.001 | p < 0.001 | |
| Spontaneous | Reference | Reference | Reference | Reference |
| Instrumental | 1.9 (1.0–3.6) | **2.1 (1.1–4.1)** | 1.6 (0.6–4.3) | 2.1 (0.9–6.1) |

*(Continued)*

**Table 2.** (Continued)

| | Moderate PPH[1] | | Severe PPH[2] | |
|---|---|---|---|---|
| | Unadjusted Odds ratio (95% CI) | Adjusted[3] Odds ratio (95% CI) | Unadjusted Odds ratio (95% CI) | Adjusted[3] Odds ratio (95% CI) |
| Caesarean section | **4.8 (4.0–5.7)** | **5.4 (4.5–6.6)** | **2.9 (2.2–3.8)** | **3.9 (2.9–5.3)** |
| Vaginal laceration | | | | |
| None or 1st grade | Reference | | Reference | |
| 2nd grade or higher | 0.8 (0.6–1.1) | | 1.4 (0.9–2.4) | |
| Birthweight (grams) | p < 0.001 | p < 0.001 | p < 0.001 | p = 0.001 |
| < 2,500 | 0.8 (0.7–1.1) | 0.2 (0.4–0.9) | **1.8 (1.3–2.6)** | 0.7 (0.4–1.1) |
| 2,500–3,999 | Reference | Reference | Reference | Reference |
| ≥ 4,000 | **2.1 (1.5–3.1)** | **1.9 (1.3–2.9)** | **2.8 (1.6–5.0)** | **2.8 (1.5–5.0)** |

[1] Blood loss 500–999 ml.

[2] Blood loss ≥ 1,000 ml, or blood loss < 1,000 ml with hemodynamic instability or three or more units blood transfusion.

[3] Adjusted: multivariate analysis of risk factors with p < 0.10 in univariate analysis and a priori risk factors (multiple gestations, parity).

[4] Ethnicity other: mixed, Chinese, Brazilian, Caucasian, or unknown.

[5] Hemoglobin ≤ 100 g/l or 6.1 mmol/l.

eclampsia in 2.6% (n = 5/194) of cases. Of the women with severe PPH, 17.1% (n = 33/193 [missing 27]) were admitted to the ICU. Among women with a CS and severe PPH (n = 89), the CS was considered elective for 53.9% (n = 48), emergency for 32.6% (n = 29), and unclassified for 13% (n = 12). Women with severe PPH had a stillbirth in 9.1% of cases (n = 20/220) in contrast to 1.6% (n = 128/7,939) stillbirth prevalence in women without PPH. Women with severe PPH and stillbirth were often diagnosed with placental abruption (85%, n = 17/20) [concomitant preeclampsia existed in 70.6% (n = 12/17) of women with placental abruption].

The management of PPH was evaluated using the criteria of the national guideline (Fig 3). AMTSL by administering oxytocin immediately after delivery was applied in 61.3% (n = 95/155 [missing 65]) of women with severe PPH. Two of the three cases of severe PPH received oxytocin treatment (68.8%, n = 106/154 [missing 66]). Of the remaining 48 women with severe PPH, 31 received no specific treatment and 17 received combined treatment with misoprostol (n = 12), methylergometrin (n = 1), and vaginal tamponade (n = 7). Tranexamic acid was administered to 4.9% (n = 5/103 [missing 117]) of women with severe PPH. While five women with blood loss below 1 liter received 6 to 10 units of blood products, eight women with blood loss ≥1,500 ml (17.4%, n = 46) received no blood products. These eight women had hemoglobin levels of at least 100 g/l and were hemodynamically stable. Blood loss was weakly to moderately correlated with the number of blood units transfused (Pearson's coefficient 0.47, p<0.01) but not with ICU admission (Pearson's coefficient 0.05, p = 0.46).

## Discussion

Based on national registry data in 2017, the prevalence rates of PPH and severe PPH in Suriname were 9.2% and 2.5%, respectively, with substantial variation across the different hospitals. Risk indicators associated with severe PPH were (1) being of African descent, (2) having a multiple pregnancy, (3) delivery in Hospital D, (4) CS, (5) stillbirth, (6) preterm birth, and (7) macrosomia. Severe PPH was mainly due to uterine atony and retained placental tissue. The criteria-based audit identified inadequate administration of oxytocin for PPH prevention (AMTSL) and therapy and infrequent use of tranexamic acid for treatment. While CS was a major risk factor, fewer women who delivered by CS received prophylactic oxytocin than women delivering vaginally did.

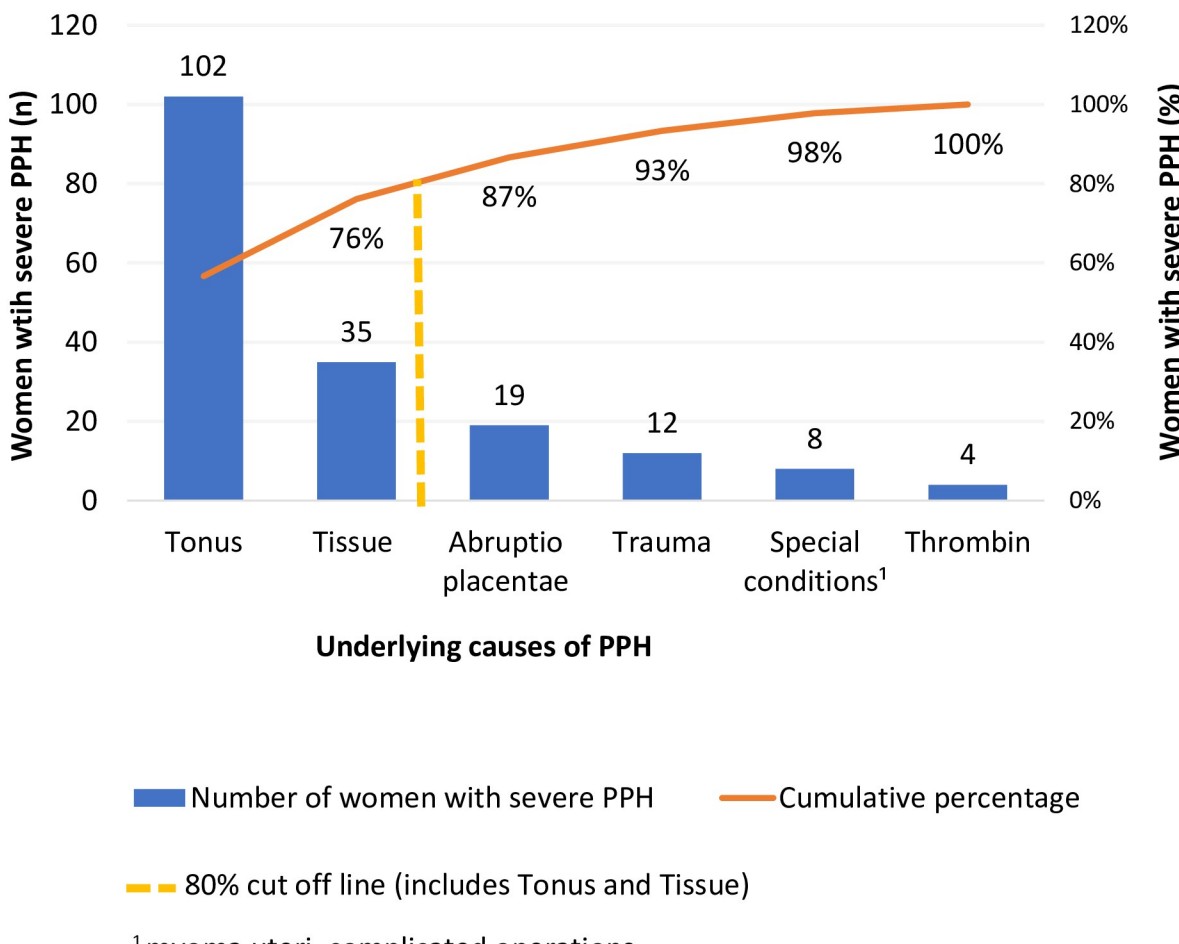

**Fig 2. Pareto chart of the specific underlying causes of postpartum hemorrhage (PPH) in Suriname in 2017.**

Worldwide and in Latin America and the Caribbean (including Suriname), PPH was the most frequent direct underlying cause of maternal deaths in 2010 [1, 17]. To reduce preventable maternal mortality from PPH in the Americas The Pan-American Health Organization (PAHO) and its Latin American Center for Perinatology, Women and Reproductive Health launched the "Zero Maternal Mortality from Hemorrhage" initiative in 2015 [24]. Following the designation of Suriname as one of 10 priority countries for reducing maternal mortality, PAHO implemented this project in Suriname in 2018 [25]. In Suriname, efforts to reduce preventable maternal deaths from PPH resulted in national PPH guideline development and obstetric emergency training in 2016 and 2019 [18, 26].

The prevalence of (severe) PPH in Suriname in this study was consistent with global and regional prevalence [10, 11]. Interhospital prevalence varied significantly despite the close geographic vicinity of four hospitals in the capital city. One explanation for this variation could be the differences in maternal, perinatal, and delivery characteristics among the hospitals as reported in this study. In Hospital D, for example, PPH prevalence was the highest, with the second-highest CS rate and higher prevalence of preterm delivery and multiparity. Another explanation for the varied interhospital PPH prevalence was the inconsistent and subjective way of obtaining information on blood loss postpartum as described previously in the methods of this study [18]. Subjective determination of the quantity of blood loss was inaccurate since blood loss

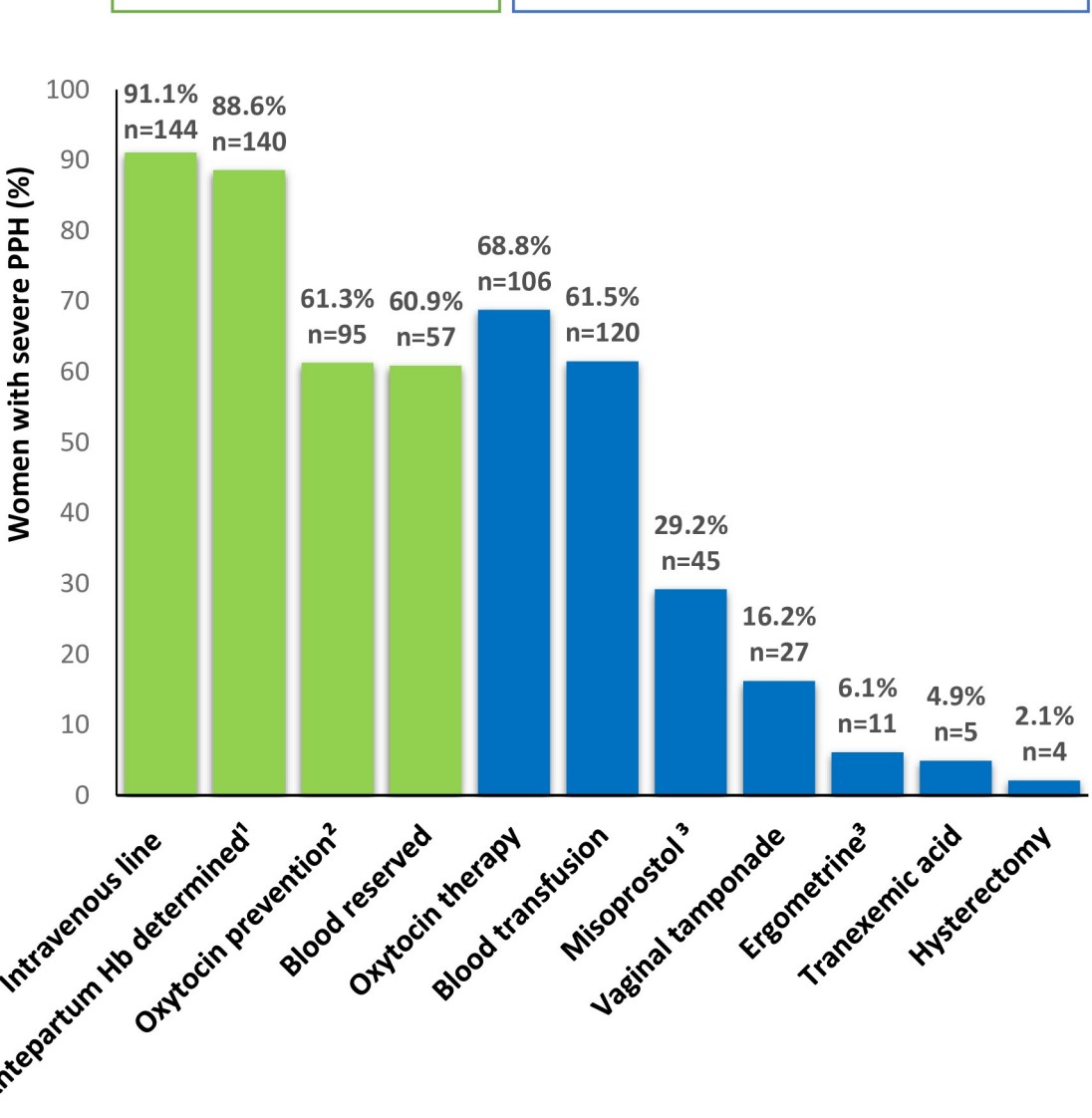

**Legend**

[1] Recent antepartum Hemoglobin (Hb) level determined

[2] Oxytocin as part of the active management of the third stage of labor (AMTSL)

[3] Ergometrine is a potent uterotonic drug and misoprostol a synthetic prostaglandin analog for PPH treatment

**Fig 3. Criteria-based audit of the management of severe postpartum hemorrhage (PPH) in Suriname in 2017 conform to the national PPH guidelines.**

was often underestimated at large volumes and overestimated at lower volumes [16, 27]. The inaccurate estimation could also explain the weak or moderate correlation between blood loss volume and PPH severity in this study. Finally, interhospital differences in PPH could result from the unequal availability of or adherence to local and national protocols and consequently, different PPH management [18]. For example, AMTSL for PPH prevention was applied less frequently in Hospital D (studied only for severe PPH). We, therefore, recommend consistent and accurate blood loss quantification conform to the PPH national guideline. During the development of this guideline health care providers reached consensus on the uniform measurement of blood loss by using a measuring cup (in mL) or weighing the pads (grams = mL). The multidisciplinary team of health care providers agreed that overestimation of blood loss could easily be prevented by changing the pads or bedpan immediately after delivery of the child [18].

The risk indicators found in this study (ethnicity, premature delivery, stillbirth, multiple gestations, CS, and macrosomia) were congruent with those reported elsewhere [28–30]. African descendants have higher risks of developing PPH compared to women of other ethnicities [31–33]. Additionally, women of African origin are more commonly anemic antepartum [34], putting them at higher risk of adverse clinical outcome when PPH develops [16]. Our study shows that women with higher antepartum hemoglobin levels stayed hemodynamically stable despite severe blood loss. This highlights the importance of prevention and treatment of antenatal anemia by routine iron and folate supplementation, especially in women of African descent [35]. Interventions aimed at preventing antenatal anemia include health and nutrition education, control of parasitic infection and improved sanitation [35]. We advise authorities to promote nationwide health education programs and campaigns to tackle antepartum anemia.

We found a strong association between severe PPH and stillbirths, which is most likely attributable to a high frequency of placental abruption among these women. A recent stillbirth study in Suriname (2016–2017) reported that placental abruption contributed to 23% of stillbirths [36]. This indicates that placental abruption and maternal conditions such as preeclampsia could be confounders in the association of stillbirth with PPH. As such, improved management of preeclampsia should reduce the risk of PPH from placental abruption.

Most healthcare workers were familiar with grand multiparity as a risk indicator and anticipated accordingly, which may explain the non-significant result found in our study. At least one risk indicator was present in most deliveries complicated by PPH but also in two-thirds of uncomplicated pregnancies without PPH. This weak discriminative ability of risk indicators to identify women who could develop PPH was also reported elsewhere [12, 14, 15]. Therefore, although risk indicator analysis should be considered to anticipate PPH occurrence, PPH can occur unforeseen, and other approaches are also needed for appropriate management.

Extrapolating the Pareto principle to our study (the "80–20 rule"), a focus on prevention of uterine atony and retained placental tissue could significantly reduce severe PPH. In AMTSL, the best preventive measure for PPH was the administration of uterotonics (oxytocin) immediately after every birth [8, 37]. The criteria-based audit showed inadequate use of prophylactic oxytocin in severe PPH. This indicates that AMTSL was not yet routine practice in Suriname in 2017 despite advice from the World Health Organization (WHO) and national guidelines [8, 18]. Tranexamic acid is an antifibrinolytic agent widely used to prevent and treat hemorrhages [38]. While sporadically used to treat severe PPH in Suriname in 2017, it is now routine practice in the first response to PPH [38]. In 2017, the WHO updated the PPH guidelines by adding the use of tranexamic acid in early PPH as advised by the World Maternal Antifibrinolytic (WOMAN) trial, which was a large multi-country randomized control trial [39, 40]. We recommend the application of AMTSL in every delivery and integration of tranexamic acid as a component of the primary treatment of PPH consistent with recent international guidelines [40, 41].

## Strengths and limitations

The strengths of this study included its national coverage. Information was obtained on the prevalence of PPH in Suriname for the first time by incorporating routinely available information from the parturition book. Application of the criteria-based audit based on national guidelines allowed for in-depth analysis of specific gaps in care to guide the prioritization of actions to reduce PPH.

This study had several limitations. First, parturition books only include facility-based deliveries or postpartum referrals, and the inclusion of primary care and home births could have resulted in lower PPH prevalence rates since the 14% primary care and home births were excluded from the analysis. The second limitation is that only *postpartum* hemorrhage was evaluated, while obstetric hemorrhage leading to mortality and severe morbidity could also result from antepartum, post-abortion, and late miscarriages. The third limitation was the higher percentage of missing data for women who delivered by CS or preterm, which are two significant risk indicators for PPH. Also, the non-significant result of the regression analysis for the risk indicators antenatal anemia, the onset of labor and perineal/vaginal laceration was probably caused by the high percentage of missing information. The prevalence of these indicators might therefore not be representative for the whole country. The fourth limitation was that several known risk factors (such as socioeconomic status, body mass index, medical history, complications in the current pregnancy, CS indication, and labor duration) could not be included in the regression analysis since these data were not available. This may explain certain observations, such as why Hospital E with the highest CS prevalence had the lowest prevalence of PPH. Finally, missing information on the causes and management of PPH impacted the criteria-based audit analysis, such as undocumented information on PPH prevention (AMTSL) among births without PPH.

## Conclusions and recommendations

Although PPH prevalence and risk indicators for Suriname are consistent with global and regional figures, wide interhospital variations exist. Since uterine atony and retained placenta are associated with almost 80% of severe PPH, intervention efforts should focus on adequate prevention, anticipation, early recognition, and prompt treatment. PPH in Suriname can be reduced by 1) prevention of PPH by applying AMTSL in every delivery and anticipating risk factors, 2) early recognition of PPH by precisely and consistently measuring blood loss, and 3) adequate therapy conforming to national guidelines. Accurate, relevant, and comprehensive data collection is essential to identify specific risk indicators and evaluate guideline implementation in the future. To gain precise insight into the gaps in PPH management, we suggest that countries focus on disaggregated data analysis and criteria-based audits.

## Supporting information

**S1 File.**
(DOCX)

## Acknowledgments

We are very grateful that Affette McCaw-Binns edited and reviewed an advanced version of this manuscript. We thank the Ministry of Health, Bureau of Public Health, and directors of the hospitals for the opportunity to collect the data. We would like to thank the registry staff of the five hospitals and individuals who contributed voluntarily to data acquisition, especially Raez Paidin, Rubinah Paidin, Raïz Boerleider, Sarah Samijadi, Sheran Henar, Shailesh

Goeptar, Clyde Moehamatdjalil, Nicole Schenkelaars, Rosemarijn Ettema, Inge Zeilstra, Stephanie Thierens, Eva van der Linden, Janine Martens, and Nienke Krijnen.

## Author Contributions

**Conceptualization:** Lachmi R. Kodan, Marcus J. Rijken, Antoon W. Grunberg.

**Data curation:** Lachmi R. Kodan, Kim J. C. Verschueren, Zita D. Prüst, Nicolaas P. A. Zuithoff.

**Formal analysis:** Lachmi R. Kodan, Nicolaas P. A. Zuithoff.

**Methodology:** Lachmi R. Kodan, Kim J. C. Verschueren, Zita D. Prüst, Nicolaas P. A. Zuithoff, Marcus J. Rijken, Antoon W. Grunberg.

**Supervision:** Marcus J. Rijken, Kerstin Klipstein-Grobusch, Antoon W. Grunberg.

**Validation:** Marcus J. Rijken, Joyce L. Browne, Kerstin Klipstein-Grobusch, Kitty W. M. Bloemenkamp, Antoon W. Grunberg.

**Writing – original draft:** Lachmi R. Kodan.

**Writing – review & editing:** Lachmi R. Kodan, Kim J. C. Verschueren, Zita D. Prüst, Nicolaas P. A. Zuithoff, Marcus J. Rijken, Joyce L. Browne, Kerstin Klipstein-Grobusch, Kitty W. M. Bloemenkamp, Antoon W. Grunberg.

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
