## [Decision Letter · Decision Letter 0]

16 Oct 2020

PONE-D-20-24164

Postpartum hemorrhage in Suriname: A national descriptive study of hospital births and an audit of case management

PLOS ONE

Dear Dr. Kodan,

Thank you for submitting your manuscript to PLOS ONE. After careful consideration, we feel that it has merit but does not fully meet PLOS ONE’s publication criteria as it currently stands. Therefore, we invite you to submit a revised version of the manuscript that addresses the points raised during the review process.

We look forward to receiving your revised manuscript.

Kind regards,

Sara Ornaghi, M.D., Ph.D.

Academic Editor

PLOS ONE

Journal Requirements:

"We also thank the UMC Utrecht Global Health Support Program for supporting LK in the research on maternal mortality in Suriname."

"The author(s) received no specific funding for this work"

Reviewers' comments:

Reviewer's Responses to Questions

**Comments to the Author**

1. Is the manuscript technically sound, and do the data support the conclusions?

Reviewer #1: Yes

Reviewer #2: Yes

2. Has the statistical analysis been performed appropriately and rigorously? 

Reviewer #1: Yes

Reviewer #2: Yes

3. Have the authors made all data underlying the findings in their manuscript fully available?

Reviewer #1: Yes

Reviewer #2: Yes

4. Is the manuscript presented in an intelligible fashion and written in standard English?

Reviewer #1: Yes

Reviewer #2: Yes

5. Review Comments to the Author

Reviewer #1: The authors sought to perform a nationwide retrospective descriptive study of all hospital deliveries in Suriname in 2017, aiming to study the prevalence, risk indicators, causes, and management of PPH to identify opportunities for PPH reduction.

This original article is well written and clear in explaining the points of strength and limits of PPH management in Suriname, mainly due to heterogeneity in management in different hospitals.

I appreciated that in the Discussion the authors described and try to find an explanation to the major issues (blood loss only visually estimated or using a jug, anemia in 36.5% of severe PPH, only 4% use of tranexamic acid).

I would suggest to expand on how increasing the accuracy in estimating blood loss (please make suggestions on which instruments and how to introduce them in the clinical scenario).

I suggest also to expand on how to prevent anemia in pregnancy (for example you should propose a campaign using posters, etc)

Reviewer #2: This is a very interesting manuscript about PPH in Suriname using national data. I only have some few comments.

Lines 70-85: The information in here must be in the introduction. Not in methods.

Line 90: please explain how jugs are used to measured blood loss after a visual estimative postpartum bleeding.

Line 119: You don’t need to explain again what is CI: confidence interval

Line 152: You have identified the hospitals using letters. It is ok, however I suggest you could include some more information, likewise the region where these hospitals are.

Table 1 – I suggest including the p-value comparing the variables.

Line 208-209 – this is result. Not discussion.

I also suggest you include in the discussion:

- antenatal anemia – do you think the prevalence is hyper estimated due to the high number of missings?

- Please discuss about the high number of missings in onset of labor and lacerations and how they could interfere results.

- Why fewer percentage of oxytocin prophylaxis when performed c-section.

- Two of the three cases of severe PPH received oxytocin 192 treatment (68.8%, n= 106/154 – one third of pph cases receive what kind of treatment?

6. PLOS authors have the option to publish the peer review history of their article (what does this mean?). If published, this will include your full peer review and any attached files.

Reviewer #1: **Yes: **Annalisa Inversetti

Reviewer #2: **Yes: **Anderson Borovac-Pinheiro

---

## [Author Response · Author response to Decision Letter 0]

29 Oct 2020

October 25th, 2020

Dear editor and reviewers,

Thank you for reviewing the manuscript " Postpartum hemorrhage in Suriname: A national descriptive study of hospital births and an audit of case management" (PONE-D-20-24164) and considering it potentially acceptable for publication. We appreciated the feedback, as suggestions helped to improve our manuscript. Kindly see below our point-by-point response to comments (in orange) and attached the revised version of our manuscript with track changes.

We hope on a positive response to the revisions we made and are looking forward to publication of our manuscript.

Yours sincerely,

On behalf of all co-authors,

Lachmi Kodan 

Reviewer #1: 

The authors sought to perform a nationwide retrospective descriptive study of all hospital deliveries in Suriname in 2017, aiming to study the prevalence, risk indicators, causes, and management of PPH to identify opportunities for PPH reduction.

This original article is well written and clear in explaining the points of strength and limits of PPH management in Suriname, mainly due to heterogeneity in management in different hospitals.

I appreciated that in the Discussion the authors described and try to find an explanation to the major issues (blood loss only visually estimated or using a jug, anemia in 36.5% of severe PPH, only 4% use of tranexamic acid).

Suggestions:

Reply: Thank you for your positive review of our manuscript, which we greatly appreciate.

I would suggest to expand on how increasing the accuracy in estimating blood loss (please make suggestions on which instruments and how to introduce them in the clinical scenario).

Reply: Thank you for your suggestion, which we considered in the Discussion where we added in line 249-253 “During the development of this guideline health care providers reached consensus on uniform measurement of blood loss by using a measuring cup (in mL) or weighing the pads (grams=mL). The multidisciplinary team of health care providers agreed that overestimation of blood loss could easily be prevented by changing the pads or bedpan immediately after delivery of the child [21].“ 

I suggest also to expand on how to prevent anemia in pregnancy (for example you should propose a campaign using posters, etc)

Reply: In the discussion in line 263-266 we added “Interventions aimed at preventing antenatal anemia include health and nutrition education, control of parasitic infection and improved sanitation [36]. We advise authorities to promote nationwide health education programs and campaigns to tackle antepartum anemia.”

Reviewer #2: 

This is a very interesting manuscript about PPH in Suriname using national data. I only have some few comments.

Reply: Thank you for reviewing our manuscript and for the comments

Lines 70-85: The information in here must be in the introduction. Not in methods.

Reply: Thank you for your comment. We moved part of this section lines 81 - 85 to the introduction (lines 61 - 67). The part describing the study setting in Suriname (lines 70-81) is left in the methods according to the STROBE guidelines (now lines 77-88).

Line 90: please explain how jugs are used to measured blood loss after a visual estimative postpartum bleeding.

Reply: To clarify how blood loss was measured we added in lines 96-97 the sentence “In case of estimated high blood loss, blood in the bedpan was measured with a measuring jug (mL) or the pads were weighted (grams=mL).”

Line 119: You don’t need to explain again what is CI: confidence interval

Reply: The explanation of the term “confidence interval” was removed from this sentence. 

Line 152: You have identified the hospitals using letters. It is ok, however I suggest you could include some more information, likewise the region where these hospitals are.

Reply: In lines 85-86 is mentioned “Four out of five major hospitals are in the capital Paramaribo; one is located at the western border of Suriname (Nickerie).” To ensure anonymity we decided not to include names of hospitals. Suriname has a relatively small population with only five hospitals where 86% of the deliveries take place. By including the region, it can easily be deducted which hospital is involved.

Table 1 – I suggest including the p-value comparing the variables.

Reply: We added the p-values in table 1 (see below). However, table 1 was not intended to describe associations between the variables and PPH. We used logistic regression analysis to compare variables and investigate the independent association of risk indicators with moderate and severe PPH. In table 2 we describe the results of the univariate and bivariate logistic regression analysis. The univariate regression analysis gives more insight in the specific category where an association exist between the variables and (moderate/severe) PPH, which is more accurate in comparing the variables than the chi-square test p-values. Therefore, we think that it was not necessary to also include p-values in table 1. We included the table with the p-values here below in case the editor still wants to add the p-values.

Table 1. Maternal, perinatal, and delivery characteristics of births in Suriname in 2017 with and without postpartum hemorrhage and undocumented blood loss. (See document "PPH response to reviewers")

Line 208-209 – this is result. Not discussion.

Reply: We removed the sentence from the discussion. This result is mentioned in lines 180-181 in the results section. 

I also suggest you include in the discussion:

- antenatal anemia – do you think the prevalence is hyper estimated due to the high number of missings?

Reply: Thank you for your comment and yes, this is possible since only data of two of the five hospitals was available, as stated in line 150 in the result section. In the limitations section in line 311-315 the possible influence of these missings are mentioned. 

- Please discuss about the high number of missings in onset of labor and lacerations and how they could interfere results.

Reply: In the limitations section in the lines 311-315 we added the sentence “Also, the non-significant result of the regression analysis for the risk indicators antenatal anemia, onset of labor and perineal/vaginal laceration was probably caused by the high prevalence of missing information. The prevalence of these indicators might therefore not be representative for the whole country.”

- Why fewer percentage of oxytocin prophylaxis when performed c-section.

Reply: We reconsidered this and decided not to mention this in the discussion and in the results section since the number of missings is relatively high (n=42/89) for CS. 

Therefore, in the discussion we deleted the lines 286-288:” especially among women delivering by CS. In contrast, according to previous interviews with healthcare providers in Suriname AMTSL was applied in all births by CS [21]” and in the results we deleted the lines 197-199 “When a CS was performed, fewer women (55.3%, n= 26/47 [missing 42]) received prophylactic oxytocin compared to vaginal births (65.1%, n= 69/106 [missing 25]).” 

-Two of the three cases of severe PPH received oxytocin 192 treatment (68.8%, n= 106/154 – one third of pph cases receive what kind of treatment?

Reply: In the current study, two of the three cases of severe PPH received oxytocin treatment (68.8%, n=106/154 [missing 66]). We added in the results in lines 200-202: “Of the remaining 48 women experiencing severe PPH, 31 received no specific treatment and 17 received combined treatment with misoprostol (n=12), methylergometrin (n=1), and vaginal tamponade (n=7).”

---

## [Decision Letter · Decision Letter 1]

3 Dec 2020

Postpartum hemorrhage in Suriname: A national descriptive study of hospital births and an audit of case management

PONE-D-20-24164R1

Dear Dr. Kodan,

We’re pleased to inform you that your manuscript has been judged scientifically suitable for publication and will be formally accepted for publication once it meets all outstanding technical requirements.

Kind regards,

Sara Ornaghi, M.D., Ph.D.

Academic Editor

PLOS ONE

Additional Editor Comments (optional):

Reviewers' comments:

Reviewer's Responses to Questions

**Comments to the Author**

1. If the authors have adequately addressed your comments raised in a previous round of review and you feel that this manuscript is now acceptable for publication, you may indicate that here to bypass the “Comments to the Author” section, enter your conflict of interest statement in the “Confidential to Editor” section, and submit your "Accept" recommendation.

Reviewer #1: All comments have been addressed

Reviewer #2: All comments have been addressed

2. Is the manuscript technically sound, and do the data support the conclusions?

Reviewer #1: Yes

Reviewer #2: Yes

3. Has the statistical analysis been performed appropriately and rigorously? 

Reviewer #1: Yes

Reviewer #2: Yes

4. Have the authors made all data underlying the findings in their manuscript fully available?

Reviewer #1: Yes

Reviewer #2: Yes

5. Is the manuscript presented in an intelligible fashion and written in standard English?

Reviewer #1: Yes

Reviewer #2: Yes

6. Review Comments to the Author

Reviewer #1: (No Response)

Reviewer #2: Congratulations for the manuscript. All the suggestions were included or justified. It is now suitable for publication.

7. PLOS authors have the option to publish the peer review history of their article (what does this mean?). If published, this will include your full peer review and any attached files.

Reviewer #1: **Yes: **Annalisa Inversetti

Reviewer #2: No

---

## [Editor Report · Acceptance letter]

9 Dec 2020

PONE-D-20-24164R1 

Postpartum hemorrhage in Suriname: A national descriptive study of hospital births and an audit of case management 

Dear Dr. Kodan:

I'm pleased to inform you that your manuscript has been deemed suitable for publication in PLOS ONE. Congratulations! Your manuscript is now with our production department. 

Kind regards, 

on behalf of

Dr. Sara Ornaghi 

Academic Editor

PLOS ONE